# Aromatic Profile, Physicochemical and Sensory Traits of Dry-Fermented Sausages Produced without Nitrites Using Pork from Krškopolje Pig Reared in Organic and Conventional Husbandry

**DOI:** 10.3390/ani9020055

**Published:** 2019-02-12

**Authors:** Martin Škrlep, Marjeta Čandek-Potokar, Nina Batorek-Lukač, Urška Tomažin, Mónica Flores

**Affiliations:** 1Agricultural Institute of Slovenia, 1000 Ljubljana, Slovenia; martin.skrlep@kis.si (M.Š.); nina.batorek@kis.si (N.B.-L.); urska.tomazin@kis.si (U.T.); 2Faculty of Agriculture and Life Sciences, University of Maribor, 2311 Hoče, Slovenia; 3Department of Food Science, IATA-CSIC, 46980 Paterna (Valencia), Spain; mflores@iata.csic.es

**Keywords:** pig, dry-fermented sausage, physicochemical, sensory characteristics, production system

## Abstract

**Simple Summary:**

Consumers associate product quality more and more with the extrinsic cues related to the way animals are raised (animal welfare), agrobiodiversity, and tradition. Consumers also favor the reduced use of additives in products. To support the preservation of the autochthonous pig breeds, which are used marginally due to their lower productivity, it is important to enhance the market potential and value of their products. In light of consumer preferences for organic farming and product naturalness, the present study was designed to develop a nitrite-free product (salami type of dry-fermented sausage) from Krškopolje pigs (autochthonous Slovenian breed) and to evaluate if and how the husbandry system (organic or conventional) affects its quality. Results of this study demonstrated softer texture and somewhat less tasty dry-fermented sausages from pigs that were held in organic husbandry. This result could be ascribed to more unsaturated fat and the fact that sausages were produced without additives with antioxidant capacity.

**Abstract:**

Dry-fermented sausages were produced in a traditional way, without addition of nitrites and starter cultures, from meat of an autochthonous breed (Krškopolje pig) raised either in a conventional indoor or organic husbandry system. Physicochemical and sensory analyses were performed at the end of processing to characterize their quality. Dry-fermented sausages from organic pork retained more moisture, which resulted in higher water activity and softer texture (instrumental and sensory). They were more oxidized (higher thiobarbituric acid reactive substances (TBARS)), in agreement with more unsaturated fatty acid profile, a higher score for rancid taste, and a higher relative abundance of volatiles from lipid β-oxidation. Overall, dry-fermented sausages from organic pork had lower levels of volatile compounds, particularly, those originating from spices (despite the same quantity added) and lower levels of amino-acid degradation. Sensory analysis showed that dry-fermented sausages from organic pork had less intensive and vivid color, tasted more bitter and sour, and had more off-tastes. The observed differences could be related to initial differences in raw material (differences in meat pH and level of polyunsaturated fatty acids) affecting the process of fermentation.

## 1. Introduction

Consumption of traditional food is driven by the positive attitudes of consumers for such products, associating it (among others) with naturalness [1]. The use of preservatives in meat dry-curing, particularly, the nitrites is important because of their role in typical color formation (stable cured color), characteristic cured aroma, microbiological safety, and oxidative stability, but also their interaction with numerous biochemical processes [2]. However, consumer concerns related to health risks associated with consumption of products containing nitrite and nitroso-derivatives [3] encourage meat processors to look for nitrite alternatives [4]. Processing of dry-fermented sausages is based on drying, which reduces the water activity, along with fermentation (under controlled temperature and relative humidity), during which the growth of lactic acid bacteria results in pH decline important for the control of spoilage bacteria and pathogens [5]. Thus, the product is stabilized for prolonged conservation. Complex physical-chemical changes occur during processing and involve major components (proteins, lipids) and additives (spices, condiments, preservation agents, microbial cultures). These changes depend on intrinsic raw material properties, microflora, and processing conditions, further affecting dehydration, microbial growth, oxidation, and which, together with the activity of meat endogenous enzymes, determine the development of final sensory characteristics [6,7]. Local (autochthonous) pig breeds are less performant and fattier than genetically improved pig breeds, which is the reason why many of them were abandoned. Preservation of these breeds as genetic resource can be assisted by developing products with attributes important for consumers (sensory quality, naturalness, and extrinsic cues associated with the way pigs are raised). In European Union, the autochthonous breeds are often used for the production of high-quality value-added products, in particular dry-cured pork products, as witnessed by their geographical or traditional specialty protection [8]. Krškopolje pig is the only Slovenian autochthonous pig breed, which despite growing interest remains endangered and used on small-scale, non-intensive farms, very often on ecological and agro-touristic farms [9], and their products are mainly made on the farm and marketed through direct sales. To improve the sustainability of the breed and its preservation, it is important to enhance market potential of pork products. Considering the consumer preferences for organic farming and product naturalness, the present study was designed to develop a nitrite-free product (salami type of dry-fermented sausage) from Krškopolje pigs and to evaluate if and how the husbandry system (organic or conventional) affects physiochemical properties, sensory traits, and volatile profile of dry-fermented sausages.

## 2. Materials and Methods 

This work was undertaken within the normal running of the farm (respecting the Slovenian law on animal protection) and no procedures were made on pigs that would demand ethical protocols according to Directive 2010/63/EU (2010). Moreover, tissue (meat) samples were taken after the slaughter. Dry-fermented sausages (salami) for the present study were processed using the meat and fat of front legs from 24 barrows of the Krškopolje pig breed, which were raised either in a conventional indoor (12 barrows) or organic system (12 barrows), receiving equivalent diet (as described in Reference [10]). Briefly, organic rearing differed from the conventional one in the availability of outdoor area (100 m^2^), the organic origin of ingredients used in feed mixture, and the lack of synthetic aminoacids and supplementation with alfalfa hay. All pigs were slaughtered in the same abattoir. After overnight cooling of carcasses, subcutaneous backfat, and meat from their front legs were shipped to the salami producer and processed the same day. Meat and backfat (representing 20% of the batter) were grinded on a 10 mm diameter mincing plate. Equal amounts of dried garlic (0.6 g/kg), ground pepper (1.0 g/kg), salt (25 g/kg), and dextrose (2.6 g/kg) were added to meat batters, mixed thoroughly, and filled into collagen casings 55 mm in diameter. Neither nitrites nor starter cultures were added to the batters. The prepared fresh sausages were put into the same drying/ripening chamber, where they were processed according to the usual practice of the processor. Initially (first week), they were held at 18 °C and relative air humidity of 82–88%. Thereafter, they were held at 12–14 °C and relative air humidity of 78–85% for 60 days. Sausages from both treatment groups were processed simultaneously, i.e., under standardized conditions of the same chamber. Randomly selected sausages were tested by accredited laboratory of Veterinary Faculty, University of Ljubljana for the presence of foodborne pathogens *Staphylococcus aureus* and other coagulase positive cocci [11], *Clostridium perfringens* [12], *Listeria monocytogenes* [13], *Salmonella* spp. [14], and *Yersinia enterocolitica* (bacteriological culture test). The shelf-life of sausages was not examined. At the end of the processing, 12 sausages (six per treatment group) were randomly selected for further analyses, vacuum packed, and frozen at −80 °C (for volatile compounds analysis) or −20 °C (other analysis). 

Chemical composition and volatile profile analysis were performed (in duplicate) on 12 dry-fermented sausages (six per treatment group) pulverized in liquid nitrogen. For determining sample pH, 1 g of sample was mixed with 4 mL of distilled water and the pH value measured using Seven2Go pH meter equipped with electrode (Mettler Toledo GmbH, Schwarzenbach, Switzerland). The content of moisture, fat and proteins, index of proteolysis (as % of non-protein nitrogen to total nitrogen), and water activity (aw) were determined as previously described [15]. To assess protein and lipid oxidation, protein carbonyl content and thiobarbituric acid reactive substances (TBARS) were determined (as previously described in References [16,17], respectively). Fatty acid composition (total and free) of dry-fermented sausage samples was analyzed in an accredited laboratory (Nutricontrol, Veghel, the Netherlands). Shortly, fat was saponified and the fatty acids esterified by the addition of methanol and boron trifluoride (BF3). Capillary gas chromatography, coupled by a flame ionization detector, was used for analysis of the resulting methyl esters. After completing the analysis of total fatty acids, the procedure was repeated without BF3 to detect bound fatty acids. Free fatty acid profile was calculated from the differences between the two runs. Volatile compounds were analyzed using solid phase microextraction (SPME), gas chromatography, and mass spectrometry (as described in Reference [18]).

For instrumental texture measurements, equal parallelepipeds (1.5 × 2 × 2 cm for height, weight, and length, respectively, four from each of 12 sampled sausages) were cut from the central part of the sausages. Stress relaxation (SR) and texture profile analysis (TPA) tests were carried out with the use of TA Plus texture analyzer apparatus (Ametek Lloyd Instruments, Ltd., Bognor Regis, UK) as described in Reference [19]. In the case of SR test, samples were compressed for 25% of their original height for 90 s and a force decay coefficient calculated as (F0-F90)/F0, where F0 is the initial force recorded at maximum compression and F90 the force measured 90 s after. For TPA test, the samples were compressed twice to 50% of their original height and the parameters hardness, cohesiveness, gumminess, springiness, chewiness, and adhesiveness were calculated, based on the data from the force-time curves recorded during the compression. For both tests, the mean value of the four technical repetitions within each sausage was calculated and used for statistical comparison.

Sensory analysis was performed on 12 (six per treatment group) dry-fermented sausages using a quantitative descriptive analysis and a panel of six male and six female trained panelists. The panel members, all employed at Agricultural Institute of Slovenia, were nonsmokers, 23–57 years old. The panel was trained using different commercially available dry-cured sausages characterized by different range of texture, tastes, and maturity to familiarize with the variation in sensory properties. During three training sessions, 15 sensory descriptors were defined describing texture (juiciness, softness, crumbliness, pastiness), color (intensity, vividness), taste (off-tastes, bitterness, rancidity, sweetness, spiciness, saltiness, sourness), and smell (typical mature smell, smell intensity). The panelists were asked not to eat or drink (except water) 2 h prior to each session. During each sensory session, panel members evaluated two sausages per treatment, and they received two 3 mm thick slices per sausage. Each sausage was evaluated twice (in two rounds of sessions) by each panelist. Descriptors were scored on a 9 cm non-structured scale, anchored at both extremes (i.e., “intensive sensation” on the right and “not detected” on the left side).

Statistical analysis consisted of one-way analysis of variance with treatment group as main effect (organic or conventional pork), using GLM procedure of SAS statistical software (SAS Institute Inc., Cary, NC, USA). In the case of sensory traits, repeated measures analysis (using panelist as random effect) was conducted using procedure MIXED of SAS. The difference between means (Tukey t-test) was considered significant at *p* < 0.05.

## 3. Results

The dry-fermented sausages were analyzed for the presence of foodborne pathogens *Staphylococcus aureus* and other coagulase positive cocci, *Clostridium perfringens, Listeria monocytogenes, Salmonella* spp., and *Yersinia enterocolitica* (bacteriological culture test), and the results were negative for all the tests.

### 3.1. Physicochemical Traits

At the end of processing, dry-fermented sausages made of meat from organic pigs had lower processing losses than sausages from conventional pigs, which resulted in the higher aw and moisture content (Table 1). Sausages from organically raised pigs had also lower pH, lower salt (NaCl) content, higher TBARS, and a tendency (*p* < 0.10) for higher carbonyl content. There were no differences observed for fat and protein content or index of proteolysis. 

### 3.2. Fatty Acids Analysis

Fatty acids (FA) profile (Table 2) showed that dry-fermented sausages from organic pork contained more PUFA (*p* < 0.001) attributable to C18:2n-6, the most abundant FA in this group, but also to C18:3 n-3 and C20:4n-6. Dry-fermented sausages from organic pork contained less SFA (*p* < 0.001) and MUFA (*p* < 0.001), mainly attributable to differences in C16:0 and C18:0, the most abundant among SFA, and C18:1n-9, the most abundant MUFA, and C20:1n-9. The levels of C16:1n7 and C17:1 (both MUFA) were, however, higher in dry-fermented sausages from organic pork (*p* < 0.05). Dry-fermented sausages from organic pork exhibited also lower amounts of free fatty acids (*p* < 0.001), owing to SFA (*p* < 0.001), MUFA (*p* < 0.001) or PUFA (*p* = 0.01). The difference in free fatty acids is mostly due to C18:1n-9 (the most abundant) with 33% lower amounts observed in dry-fermented sausages from organic than conventional pork.

### 3.3. Volatiles Profile Analysis

There were 66 individual volatile compounds identified in dry-fermented sausages (Table 3). Classification of volatiles according to the most probable origin [20] showed that the most abundant were those deriving from the spices, followed by the volatiles generated by microbial carbohydrate fermentation, lipid autooxidation, esterase activity, degradation of amino acids, and lipid β-oxidation, accounting for 33.7 %, 23.3%, 13.7%, 13.2%, 10.3%, and 5.9% of relative abundance/total area, respectively.

As depicted in Figure 1, dry-fermented sausages from organic pork had lower levels (*p* = 0.017) of total volatile compounds than sausages from conventional pork. This effect was most notable for the volatiles originating from spices (*p* < 0.001) and was observed for 16 out of 20 identified substances (most markedly for limonene, sabinene, α-phellandrene, and terpenes as the most abundant compounds within this group). Dry-fermented sausages from pork of organic system were also characterized by lower (*p* = 0.008) level of volatiles generated by amino acid degradation. This was observed for several individual volatiles from this group (i.e., benzenacetaldehyde, benzenaldehyde 3-methyl butanal), while 2-methyl propanol and methional were not identified in the sausages from organic pork. With regard to the volatiles originating from lipid β-oxidation, their higher abundance (*p* = 0.006) was observed in the sausages from organic pork, the difference being mostly attributable to (R)-2-butanol. There were no differences between the sausages in volatiles originating from carbohydrate fermentation, although several individual compounds were found in either higher (i.e., acetic acid, 2-butanone) or lower (i.e., ethanol, acetone, 3-hydroxy-2-butanone) amounts in sausages from organic pigs (*p* < 0.05). Similarly, mainly no significant differences (*p* > 0.05) between dry-fermented sausages from organic and conventional pork were observed for volatiles generated by lipid autooxidation, though some individual volatiles were affected (i.e., 2-pentyl-furan and pentanal had higher, whereas nonanal had lower abundance in sausages from organic pigs, *p* < 0.05).

### 3.4. Instrumental Texture Parameters

Significant differences in some of instrumentally assessed texture properties (Table 4) were observed between dry-fermented sausages made of meat from organic or conventional pigs. The texture of dry-fermented sausages made of organic pork was softer, with lower cohesiveness, lower gumminess, and lower chewiness. 

### 3.5. Sensory Analysis

Results of sensory analysis of dry-fermented sausages from organic or conventional pork (Figure 2) also exhibited significant differences related to pig husbandry practice. The panelists scored the color of dry-fermented sausages from organic pigs as being less intensive and somewhat less vivid. The panelists perceived dry-fermented sausages from organic pigs as softer, crumblier, and juicier. As for the taste, panelists found dry-fermented sausages from organic pork as being more bitter, sourer, and slightly more rancid. Dry-fermented sausages from meat of organic pigs received also higher average score for off-tastes.

## 4. Discussion

Physical-chemical properties of dry-fermented sausages manufactured without nitrite addition differed with regard to organic or conventional husbandry origin of meat. Higher moisture content (along with higher water activity) can be ascribed to lower processing losses in the sausages from organic pork. Considering that fat content was similar in both sausage groups, lower processing losses in the case of organic sausages can be explained with fatty acid composition, namely, higher PUFA content. More unsaturated fats are prone to exudation from the adipose tissue cells, covering the meat particles inside the sausage and causing oiliness on the product surface, which prevents the product from drying [21]. Differences in pH values might be due to the meat intrinsic properties and/or microbial fermentation. It was shown [22] that spontaneous fermentation is characterized by richer microbial diversity, delayed lactic acid bacteria evolution, and lower pH decline (slow fermented sausages have generally milder pH declines). The pH value was already lower at start in organic than conventional meat mixture (6.10 vs. 6.24). However, for sausages, the meat from front legs was used (composed of more oxidative muscles with higher pH), the differences in pH were also evidenced in glycolytic *longissimus dorsi* muscle of the same pigs [10]. Different pH of meat batter at start has likely affected the development of the autochthonous microflora (no starter cultures were used) and, consequently, the fermentation process [6]. However, as no analysis of main microbiota populations were carried out, it is difficult to make any further conclusions. The observed differences in salt content between organic and conventional group are difficult to explain. Overall the difference, although significant, is small (i.e., less than 0.3% points) and could be due to low intra-group variability. Higher oxidation levels observed for organic sausages, either of protein (i.e., carbonyls) or lipids (i.e., TBARS), could also be associated with higher PUFA levels [23,24]. Although TBARS concentrations are not proportional to protein carbonyl concentrations measured by dinitrophenylhydrazine (DNPH) method (as in our case), they are, in general, positively correlated [25] because lipid oxidation is one of the main factors governing the oxidation of proteins and amino acids [23,24]. In addition, oxidation processes in dry-fermented sausages may be stimulated by several other factors including lower pH value [20], as observed in organic compared to conventional sausages. The observed effect confirms the result of our previous report [18], just that the level of oxidation was higher (by 30 and 58% for TBARS and carbonyls, respectively) due to nitrites absence, as the nitrites are potent antioxidants.

With regard to fatty acid composition, higher levels of PUFA (in parallel to lower SFA) observed in organic sausages may be attributed to the differences in the diet, namely, the supplementation with alfalfa hay, rich in polyunsaturated fats. The share of PUFA in total fatty acids of the alfalfa hay represents more than 60% [26]. Higher PUFA in fat from organic or outdoor reared pigs with access to green feed agrees with the results of other studies [27,28]. Additionally, lower fat saturation in organically raised pigs may also be due to higher physical activity, as shown for rats [29,30], but also in pigs [31], i.e., exercised Iberian pigs exhibiting lower SFA in the backfat, higher PUFA in neutral fat of *Psoas major* muscle than sedentary ones, despite being fed the same diet. Sausages from organic pork had also lower amounts of FFA than conventional ones. This may indicate lower level of lipolysis or higher level of FFA oxidative degradation (in line with higher TBARS) in organic sausages. As reviewed by Reference [21], lipolysis and lipid oxidation are not positively associated, indications exist that lipolysis even prevents FFA from oxidation, which may additionally explain the differences between organic and conventional sausages of the present study. In fermented meat products with high pH (case of conventional group), lipases from very lipolytic species of *Staphylococcus* could increase lipolysis [32]. In general, the factors affecting lipase activity are insufficiently explained, either in regard to *ante mortem* factors, like pig weight or diet [21], or in relation to meat batter formulation (additives, pH [33]). A decrease of neutral lipase activity was likely due to lower pH and may also be associated to moderate exercise of pigs [31], which could explain lower lipolysis in sausages from organic pigs. On the contrary, the activities of acid lipase and esterase are increased with lower pH, salt concentration, and aw reduction [34], which might explain higher lipolysis in conventional sausages of the present study.

Differences in volatile compounds may be attributable to microbial and/or endogenous enzymes, however, as the microbial populations were not examined in the present study, it is not possible to ascertain if and to which degree the differences in volatiles derive from microbial metabolism or other processes like auto-oxidation or endogenous enzyme activity. It can, however, be speculated that when higher concentrations of volatiles classified as products of microbial lipid β-oxidation are found, microbial growth and/or metabolism must have been stimulated by milieu conditions such as pH, salt content, and moisture [6]. Moreover, since the sausages were made without nitrites, it is likely that the differences between sausages made from organic or conventional pork were exacerbated due to the absence of antioxidant and anti-microbial function of nitrites [4,35]. As for individual volatile groups, the most notable and uniform difference between sausages from organic and conventional pork was found for volatiles originating from the added spices (namely, garlic and pepper), indicating app. three-fold reduction in organic compared to conventional sausages. One explanation may be their higher degradation in organic sausages due to more oxidative conditions, indicated by oxidation markers carbonyls and TBARS as well as higher concentrations of PUFA. Many compounds of spices (for example, terpenoids) are relatively unstable [36] and disposed to oxidation as shown, for instance, on black pepper oleoresin [37]. Initially formed oxidation products from terpenoids were also reported to decompose upon increasing acidity, while volatile stability may also be compromised by the presence of moisture [36]. In the present case, both factors were more pronounced in the sausages from organic pork. Furthermore, for garlic extracts (thiosulfanates), decreasing stability was reported with increasing degree of fatty acid unsaturation [38]. Garlic extracts were also found to be surprisingly unstable when stored in liquid vegetable oil [39], which has high PUFA. Although there was no difference in proteolysis (i.e., proteolysis index) noted between sausages from organic and conventional pork, we could observe that sausages from organic meat had lower amounts of volatiles originating from amino acid degradation. The reasons for that may be in more specific proteolytic degradation of either endogenous or microbial enzymes. In support of present study results, the research on the Iberian breed [40] demonstrated that exercised pigs, in comparison to sedentary ones, expressed lower activity of some muscle proteolytic enzymes and lower branched chain amino acids levels, but no differences in total free amino acids. Branched chain amino acids act as precursors for Strecker aldehydes which contribute to flavor of dry-fermented sausages [41]. One (3-methyl-butanal) was detected at lower levels in sausages from organic pork. A similar result was observed for the most abundant volatile compound from this group, benzenacetaldehyde, originating from Strecker degradation of phenylalanine [42]. The Strecker degradation is favored by low aw [7] and the higher concentrations of precursor free amino acids (generated by aminopeptidases, enzymes with lower activity at lower pH [6]) which explains lower levels amino acid-derived volatiles in sausages from organic pork.

Instrumental texture differences match closely the moisture content, which can be explained by fat unsaturation and its effect on dehydration process. Less saturated fat with its softer consistency [43] could also have directly affected the softer texture of sausages from organically raised pigs.

Regarding sensory analysis, color intensity (lower in sausages from organic pork), juiciness, softness, and crumbliness (higher in sausages from organic pork) corroborate with higher moisture content. Lower color intensity and vividness of sausages from organic pork can be associated to higher level of oxidation, namely, oxidative processes of lipids associated to the formation of yellow colored polymers [44] and discolorations due to the oxidation of muscle pigment myoglobin that can be enhanced either by lipid oxidation products or directly by lower pH [45]. Higher sourness of sausages from organic pork can be related to either higher acetic acid content or lower pH value. Higher off-taste scores, along with higher rancidity, were observed in sausages from organic pork, which were more prone to oxidation and agrees with higher TBARS and carbonyl concentrations. Although there was no significant difference in smell intensity or typicity, we could note lower total volatile compounds (VOCs) along with higher score for (negative) taste attributes, such as bitterness, sourness, rancidness, and off-tastes in sausages from organic pork. Higher off-taste scores in sausages from organic pork can be aligned with lower levels of volatiles originating from spices or higher levels of other specific volatiles. This includes 2-pentyl-furan (characterized by rancid, onion, savory, stable, sulfur [7]), pentanal (characterized by strong pungent, rancid, green notes [7,46]), acetic acid and 2-butanol (characterized by vinegar, sour or winey notes [7,47]), or 2-butanone (associated to ethereal note [47]). In addition, sausages from organic pork exhibited also lower levels of volatiles characterized by more positive aroma notes, like 3-hydroxy-2-butanone (buttery, sweet caramel, yogurt [7,48,49]), 3-methyl butanal (fruity, cheesy, malty, cured, green, acorn [7,47]), acetaldehyde (green [7]), ethyl alcohol (sweet, alcohol, bread, yeast [7,47]), benzenacetaldehyde (musk, jasmine, rancid [50]), and benzaldehyde (bitter almonds, herbal, spices, pine [7]).

## 5. Conclusions

Organically reared pork was characterized by lower meat pH and higher fat unsaturation, which led to higher moisture retention and higher levels of oxidation in dry-fermented sausages, resulting in softer texture, less intensive color, and changed volatile profile (less volatiles originating from spices and amino acid degradation, more from lipid β-oxidation) and sensory quality (higher presence of sour, bitter, rancid, and off-tastes). It is possible that, in the present study, the effect was more pronounced because the sausages were produced without nitrites and starter cultures. More studies are needed with organic pork and use of natural antioxidants to reply to consumer demand for more traditional and natural products. 

## Figures and Tables

**Figure 1 animals-09-00055-f001:**
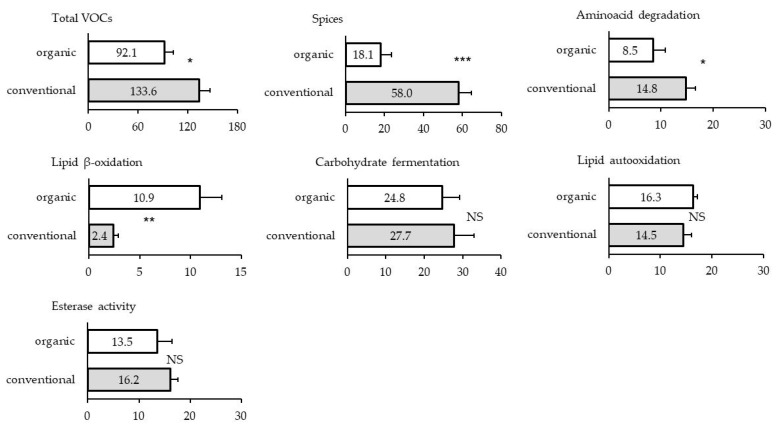
Relative abundance of groups of volatile compounds (VOCs) in dry-fermented sausages (six per treatment group) according to pig husbandry practice (organic or conventional). (NS – *p* > 0.10, * – *p* < 0.05, ** – *p* > 0.01, *** – *p* < 0.001)**.**

**Figure 2 animals-09-00055-f002:**
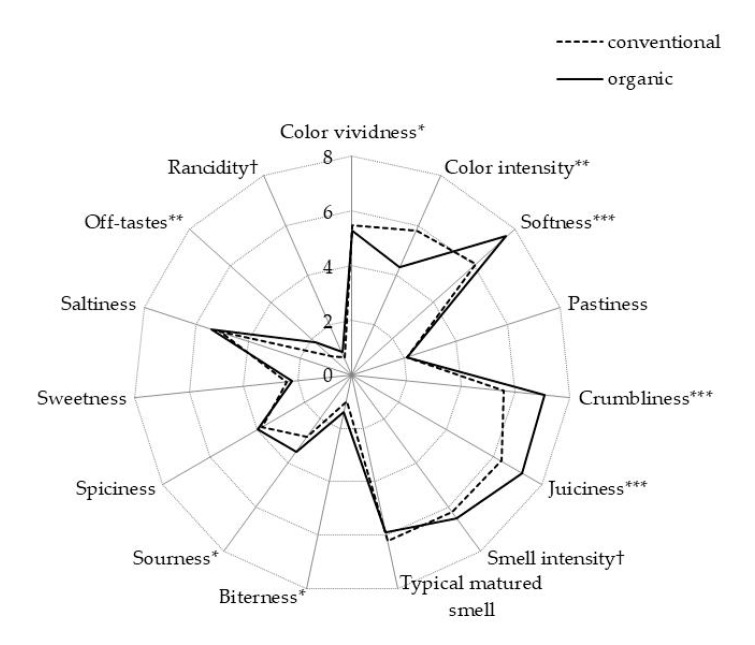
Sensory profile of dry-fermented sausages (six per treatment group) manufactured without nitrite addition from pigs fattened in conventional or organic system (NS – *p* > 0.10, * – *p* < 0.05, ** – *p* > 0.01, *** – *p* < 0.001).

**Table 1 animals-09-00055-t001:** Physicochemical traits of dry-fermented sausages (six per treatment group) manufactured without nitrite addition from pigs fattened in conventional or organic system.

Physicochemical Traits	Organic	Conventional	RMSE	Significance
Weight loss during processing, %	32.8	37.1	0.63	***
Water activity (aw)	0.910	0.868	0.0098	***
Moisture, %	31.3	26.7	0.40	***
Fat, g/kg DM	599	593	14.2	NS
Protein, g/kg DM	329	327	13.52	NS
pH	5.83	6.27	0.082	***
Proteolysis index, %	9.2	9.6	1.51	NS
NaCl, g/kg DM	64.7	67.5	1.78	*
TBARS, μg MDA/kg DM	87.8	80.5	3.17	**
Carbonyls, nmol/mg proteins	5.3	4.1	1.11	†

RMSE = Root mean square error. DM = Dry matter. TBARS = Thiobarbituric acid reactive substances. MDA = Malondialdehyde. Significance: NS – *p* > 0.10, † – *p* < 0.10, * – *p* < 0.05, ** – *p* < 0.01, *** – *p* < 0.001.

**Table 2 animals-09-00055-t002:** Fatty acid and free fatty acid composition (g/100 g fat) of dry-fermented sausages (six per treatment group) manufactured without nitrite addition from organic and conventional pork.

Fatty Acid Composition	Organic	Conventional	RMSE	Significance
Fatty acids				
C14:0	1.40	1.30	0.000	***
C16:0	24.70	25.17	0.131	**
C16:1n7	2.28	2.17	0.065	**
C17:0	0.70	0.70	0.045	NS
C17:1	0.38	0.30	0.029	**
C18:0	13.67	14.37	0.144	***
C18:1n-9	37.75	38.65	0.207	***
C18:1 other	3.08	3.07	0.079	NS
C18:2n-6	12.38	10.85	0.246	***
C18:3 n-3	0.85	0.68	0.048	***
C20:1n-9	0.80	0.92	0.029	***
C20:4n-6	0.28	0.22	0.002	**
SFA	41.00	41.98	0.291	***
MUFA	44.18	44.98	0.271	**
PUFA	14.47	12.65	0.288	***
Free fatty acids				
C14:0	0.07	0.08	0.008	**
C16:0	0.67	1.08	0.105	***
C18:0	0.25	0.37	0.054	**
C18:1n-9	1.58	2.37	0.202	***
C18:1 other	0.17	0.23	0.024	**
C18:2n-6	0.96	1.06	0.066	*
C18:3n-3	0.07	0.07	0.006	NS
C20:1n-9	nd	0.07		
C20:2n-6	0.07	0.08	0.008	*
C20:4n-6	0.01	0.03	0.028	NS
C22:3n-3	0.06	0.07	0.023	NS
C24:1n-9	0.02	0.04	0.035	NS
SFA	0.98	1.23	0.164	***
MUFA	1.90	2.86	0.232	***
PUFA	1.17	1.31	0.088	*
Total	4.05	5.70	0.476	***

SFA = Saturated fatty acids. MUFA = Monounsaturated fatty acids. PUFA = Polyunsaturated fatty acids. RMSE = Root mean square error. nd = not detected. Significance: NS – *p* > 0.10, * – *p* < 0.05, ** – *p* < 0.01, *** – *p* < 0.001.

**Table 3 animals-09-00055-t003:** Volatile compounds (expressed as normalized area (Area Comp/Area IS); IS=2-methyl-3-heptanone) of dry-fermented sausages (six per treatment group) manufactured without nitrite addition made of pork from organic or conventional husbandry system.

Supposed Origin of Volatiles	LRI	RI	Organic	Conventional	RMSE	Significance
Spices						
Allyl mercaptan	610	b	0.43	0.37	0.163	NS
Allyl methyl sulphide	718	a	4.43	5.69	1.004	†
(Z)-1-(methylthio) 1-propene	759	b	0.40	0.28	0.062	**
Allyl sulfide	883	a	0.15	0.42	0.178	*
Terpene	934	a	1.14	4.21	0.885	***
α-pinene	940	a	0.63	2.37	0.501	***
Sabinene	986	a	3.25	12.80	2.518	***
β-myrcene	1003	a	0.25	2.01	0.375	***
α-phellandrene	1022	a	1.24	8.39	1.555	***
a-terpinene	1034	a	0.19	0.76	0.159	***
Limonene	1045	a	3.70	15.09	2.902	***
Terpene	1050	b	0.92	5.04	1.442	*
γ-terpinene	1074	b	0.28	1.01	0.201	***
Terpene	1099	b	nd	0.09	/	/
Terpinolene	1101	a	0.09	0.47	0.095	***
Diallyl disulphide	1119	a	0.20	0.20	0.042	NS
1,2-dimethoxy-Benzene	1197	b	0.07	0.11	0.027	*
Toluene	788	a	0.10	0.15	0.050	NS
Copaene	1403	b	0.21	0.37	0.068	**
Caryophyllene	1434	a	0.78	1.65	0.295	***
Aminoacid degradation						
Benzaldehyde	1018	a	1.15	3.64	0.861	**
Benzeneacetaldehyde	1111	a	7.10	10.29	2.307	*
2-Methyl propanal	592	a	nd	0.08		
3-Methyl butanal	689	a	0.25	0.99	0.287	**
Methional	966	a	nd	0.10		
Lipid β-oxidation						
Isopropyl alcohol	537	a	0.24	0.39	0.094	*
(R)-2-Butanol	642	a	9.90	1.54	4.182	**
2-Pentanone	733	a	0.17	0.18	0.050	NS
1-Octen-3-ol	1031	a	0.51	nd		
2-Octanone	1039	a	0.10	0.10	0.027	NS
2-Nonanone	1140	a	0.22	0.30	0.109	NS
Carbohydrate fermentation						
Ethyl alcohol	505	a	2.66	3.86	0.715	*
Acetaldehyde	466	a	0.07	0.11	0.036	†
Acetic acid	711	a	6.93	4.80	1.374	*
Acetone	527	a	2.33	12.76	7.668	*
2-Butanone	630	a	11.51	3.57	3.287	**
3-Hydroxy-2-butanone	779	a	0.41	1.51	0.606	*
Butanoic acid	887	a	0.85	1.11	0.417	NS
Lipid autooxidation						
1-Propanol	611	a	0.76	0.93	0.532	NS
Propanal	523	a	0.09	0.07	0.034	NS
2-Pentyl-furan	1009	a	0.09	0.05	0.011	**
Pentane	500	a	2.29	2.06	0.698	NS
Pentanal	738	a	0.34	0.29	0.085	*
Hexane	600	a	0.94	1.10	1.015	NS
1-Hexanol	923	a	0.47	0.05	0.253	NS
Hexanal	840	a	5.13	4.29	1.578	NS
Heptane	700	a	2.42	2.45	0.610	NS
(Z)-2-Heptenal	1011	a	nd	0.19		
Octane	800	a	2.95	2.62	0.638	NS
Octanal	1049	a	0.19	nd		
Octanoic acid	1264	a	0.03	0.05	0.018	NS
Nonanal	1149	a	0.47	0.62	0.114	*
Esterase activity						
Methyl acetate	549	a	4.76	5.04	1.363	NS
Ethyl acetate	634	a	0.71	0.48	0.228	NS
Methyl propionate	650	a	0.89	1.31	0.763	NS
Methyl butanoate	755	a	3.19	5.00	1.821	NS
Methyl 2-hydroxypropanoate	792	a	0.20	0.13	0.047	*
Methyl 3-methylbutanoate	805	a	0.96	1.31	0.576	NS
Ethyl butanoate	830	a	0.51	0.70	0.147	NS
Methyl pentanoate	855	a	0.13	0.13	0.029	NS
Propyl butanoate	925	a	nd	0.09		
Methyl hexanoate	951	a	1.53	1.63	0.373	NS
Methyl heptanoate	1057	a	0.07	0.05	0.014	†
Methyl octanoate	1156	a	0.78	1.16	0.352	†
Methyl nonanoate	1260	a	nd	0.14		
Methyl decanoate	1358	a	0.14	0.24	0.127	NS

LRI = Linear retention index for DB-624 column. RI = Reliability of identification: a = Identification by mass spectrum and by coincidence with the LRI of an authentic standard, b = Tentative identification by mass spectrum. RMSE = Root mean square error. nd = not detected. Significance: NS – *p* > 0.10, † – *p* < 0.10, * – *p* < 0.05, ** – *p* < 0.01, *** – *p* < 0.001.

**Table 4 animals-09-00055-t004:** Instrumental texture parameters of dry-fermented sausages (six per treatment group) manufactured without nitrite addition from pigs fattened in conventional or organic system.

Texture Parameter	Conventional	Organic	RMSE	Significance
Force decay coefficient	0.57	0.60	0.066	NS
Hardness, N	57.7	30.7	7.60	***
Cohesiveness	0.51	0.38	0.054	***
Gumminess, N	29.9	11.9	5.90	***
Springiness, mm	4.4	4.5	0.68	NS
Chewiness, N	135.3	53.8	33.98	***
Adhesiveness, N*mm	−2.5	−2.9	1.10	NS

RMSE = Root mean square error. Significance: NS – *p* > 0.10, † – *p* < 0.10, * – *p* < 0.05, ** – *p* < 0.01, *** – *p* < 0.001.

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
