# Peer review of "Aromatic Profile, Physicochemical and Sensory Traits of Dry-Fermented Sausages Produced without Nitrites Using Pork from Krškopolje Pig Reared in Organic and Conventional Husbandry"

_animals, 2019, doi:10.3390/ani9020055_

Round 1

Reviewer 1 Report

After revision I think the paper can be accepted for publication

Reviewer 2 Report

Any comments for this version. 

This manuscript is a resubmission of an earlier submission. The following is a list of the peer review reports and author responses from that submission.

Round 1

Reviewer 1 Report

Dear Authors 

In my opinion, it is an interesting paper but must be improved in some aspects.

The title uses the term "Ecological" but throughout the paper uses "Organic". Please, choose one of them for both - title and text and tables.  

The objective of this study is not clearly presented. It is important to know if the objective is to compare the products obtained from conventional vs organic systems or from organic vs conventional systems. This order is important for the presentation and discussion of the results. In the text you always compare the products from organic systems with the conventional systems but in the tables the order is the reverse: 1st column refers to conventional and the 2nd to organic. In my opinion, this confuses the reader. So, I suggest the use of the same order in the text and in the tables. I also suggest the use of abbreviation to the different systems - Conventional (C), organic (O). 

Please, confirm that is correct the use of to raise as synonim of to rear or to breed.

It will be very useful to explain the differences between the two husbandry systems. We do not know them and it is important for understanding some comments in the paper related with exercise, diets etc. that may influence the products quality. 

In the tables, I suggest to indicate the significant differences with the symbols *, ** or *** , maintaining the P-value.

The order presented to describe the methodology used (Materials and Methods) and to discuss the results (Discussion) must be  followed in Results : 3.1 - Physico-chemical traits; 3.2 - Fatty acid analysis; 3.3 Volatile profile analysis; 3.4 Instrumental texture parameters ; 3.5 - Sensory analysis. This is the order you follow in "Materias and methods" and in "Discussion"

Table 5 must be improved as follows: 

Please, insert a title in the column to refer the most probable origin of volatiles (1st column) and a row with the total value of each group of VFA with the same origin. 

The title of the groups "lipid beta-oxidation" and "esterase activity" are nor evidenced, as the other titles, at the left of the column. 

Line 73 - I suggest: "...receiving equivalent diet, as described in Tomazin et al., 2018 [10].

Line 74 - I suggest : "All pigs were slaughtered in the same abattoir. After overnight cooling of carcasses...."

Line 86 - Staphylococcus aureus must be in italic. 

Line 87 - You may refer the methodology utilized for microbiological analysis of the different microrganisms. ISO? The sentence "Results were negative" must be in "Results", not in "Materials and Methods".

Line 186 - I propose that Fig 1, and the comments about it, be presented beforeTable 5. 

In Fig 1, the graphics must be presented following the same order in which they are commented. Unless a better opinion, I suggest: 1st - total VOCS; 2nd - Spices; 3rd - aa degradation; 4th - lipid beta-oxidation; 5th carbohydrate metabolism; 6th lipid autooxidation; 7th sterase activity.

The order of these results inTable 5 must be in accordance, as far as possible. 

In Fig 1 - what does mean "VOCS"? - extense, please.

Line 204 - "carbohydrate metabolism" mus be changed to carbohydrate fermentation, please.

Line 213 - The sentence must be supported by a reference, at least.

Line 234 - What does mean "DNPH"? - extense, please.

Line 246 - "but" - the initial must be in capital letter.

Lines 272 and 273 - to understand this sentence is essential to know the total value of the volatiles originating by each group. As I mentioned before, insert a row with these values in Table 5, please.

Line - 287 - A brief characterization of the husbandry systems is welcome to understand that  exercise is determinant for these differences.

Line 295 - The ) is missing

Line 309 - VOCS - extense, please. Write in full before abbreviation, please.

Line 319 - The ) is missing

Line 320 - the ) is missing.

Best regards 

Author Response

Dear Reviewer 1, 

Thank you for your useful comments and suggestions. Below please find our answers:

Q: The title uses the term "Ecological" but throughout the paper uses "Organic". Please, choose one of them for both - title and text and tables.  

A: in the title, the term "ecological" was replaced with the term »organic«

Q: The objective of this study is not clearly presented. It is important to know if the objective is to compare the products obtained from conventional vs organic systems or from organic vs conventional systems. This order is important for the presentation and discussion of the results. In the text you always compare the products from organic systems with the conventional systems but in the tables the order is the reverse: 1st column refers to conventional and the 2nd to organic. In my opinion, this confuses the reader. So, I suggest the use of the same order in the text and in the tables. I also suggest the use of abbreviation to the different systems - Conventional (C), organic (O). 

A: indeed, we compare organic to conventional (i.e. control). We changed the order in the tables. We do not use abbreviations as there is enough space to write in full, being in the tables or figures.

Q: Please, confirm that is correct the use of to raise as synonim of to rear or to breed.

A: we can confirm that the usage of »raising animals« is OK. Rear and raise are synonyms, while “to breed” refers to reproductive phase

Q: It will be very useful to explain the differences between the two husbandry systems. We do not know them and it is important for understanding some comments in the paper related with exercise, diets etc. that may influence the products quality. 

A: we added a sentence to the text, explaining the main differences between two systems (Lines 75-77).

Q: In the tables, I suggest to indicate the significant differences with the symbols *, ** or *** , maintaining the P-value.

The order presented to describe the methodology used (Materials and Methods) and to discuss the results (Discussion) must be  followed in Results : 3.1 - Physico-chemical traits; 3.2 - Fatty acid analysis; 3.3 Volatile profile analysis; 3.4 Instrumental texture parameters ; 3.5 - Sensory analysis. This is the order you follow in "Materias and methods" and in "Discussion"

A: Done as suggested.

Q: Table 5 must be improved as follows

Please, insert a title in the column to refer the most probable origin of volatiles (1st column) and a row with the total value of each group of VFA with the same origin. 

A: Done as suggested.

Q: The title of the groups "lipid beta-oxidation" and "esterase activity" are nor evidenced, as the other titles, at the left of the column. 

A: Aligned to left.

Q: Line 73 - I suggest: "...receiving equivalent diet, as described in Tomazin et al., 2018 [10].

A: Corrected as suggested (Line 74).

Q:Line 74 - I suggest : "All pigs were slaughtered in the same abattoir. After overnight cooling of carcasses...."

A: Corrected as suggested (Line 77-78).

Q:Line 86 - Staphylococcus aureus must be in italic. 

A: Corrected as suggested (Line 89).

Q:Line 87 - You may refer the methodology utilized for microbiological analysis of the different microrganisms. ISO? The sentence "Results were negative" must be in "Results", not in "Materials and Methods".

A: Methodology references (ISO standards) were added (Lines 90, 389-398), and the request to move the sentence followed (Lines 140-143).

Q:Line 186 - I propose that Fig 1, and the comments about it, be presented before Table 5. 

A: Corrected as suggested.

Q: In Fig 1, the graphics must be presented following the same order in which they are commented. Unless a better opinion, I suggest: 1st - total VOCS; 2nd - Spices; 3rd - aa degradation; 4th - lipid beta-oxidation; 5th carbohydrate metabolism; 6th lipid autooxidation; 7th sterase activity.

The order of these results in Table 5 must be in accordance, as far as possible. 

A: Corrected as suggested.

Q:In Fig 1 - what does mean "VOCS"? - extense, please.

A: VOCs denotes volatile compounds; now explained (Lines 196, 328)

Q:Line 204 - "carbohydrate metabolism" must be changed to carbohydrate fermentation, please.

A: Corrected (line 188).

Q:Line 213 - The sentence must be supported by a reference, at least.

A: We do not understand the need for the reference – this conclusion is evident from our results; i.e. lower weight losses during drying result in more (retained) moisture.

Q:Line 234 - What does mean "DNPH"? - extense, please.

A: abbreviation explained (line 252).

Q:Line 246 - "but" - the initial must be in capital letter.

A: not applicable as the dot was there by mistake; now deleted and the sentence is continued (Line 265).

Q:Lines 272 and 273 - to understand this sentence is essential to know the total value of the volatiles originating by each group. As I mentioned before, insert a row with these values in Table 5, please.

A: This information was already available in the Figure 1 but for better clarity we added the values for relative abundance of VOCs in the figure. It would be redundant to have it in the Table as well.

Q:Line - 287 - A brief characterization of the husbandry systems is welcome to understand that  exercise is determinant for these differences.

A: A brief characterisation of main differences between husbandry systems are explained now in lines 75-77.

Q:Line 295 - The ) is missing

A: Corrected

Q:Line 309 - VOCS - extense, please. Write in full before abbreviation, please.

A: Written in full before abbreviation

Q:Line 319 - The ) is missing

A: Corrected

Q:Line 320 - the ) is missing.

A: Corrected

Best regards 

Reviewer 2 Report

The draft presents an interesting subject, that can be useful for regional pork value chains. All the knowledge acquired can improve the quality of the food products and so the sustainability of the Krškopolje pig scarcely studied until now. However, in the last year, several research works about different aspects of this race have been published.

The draft is well organized and comprehensively written.

The title should be wider in order to include all the aspects considered in the work. The abstract is well written and resume all the work.

The state of the art is updated and easy to understand.

All the methods have appropriate and relevant references of other authors. The design and organization of the trial (considering the concepts of samples, repetitions, treatments, etc.)  should be deeply explained and the words systematically used.

The final discussion is supported by the results obtained during the current research work and also by the results from other relevant researchers.

The conclusions are consistent and according to the goals and previous discussion.

Detailed comments:

Line 86 - Staphylococcus aureus replace by Staphylococcus aureus

Line 88 to 90 - “At the end of the processing, sausages from each treatment group (n=6 per treatment group) were randomly selected for further analyses, vacuum packed and frozen at -20°C or -80°C (depending on the analysis)”. To explain this better, namely the number of sausages.

Is necessary to explain how many sausages for each treatment group for each analysis
.

Line 107 - How many sausages?

Line 117 - Why to use the descriptor (sensory evaluation) rancid? Maybe that kind of flavour is due to other causes like volatiles originating from spices and not really rancidity.

Line 162 - Instead of table 3 to show data of sensory evaluation I suggest a graphic representation in order to allow the reader to easier compare treatment results. The data can be plotted onto graphs, such as the spider plot.

Line 231 “The observed differences in salt content between organic and conventional group are difficult to explain”. Maybe the explanation is the higher value of weight loss during the process for conventional sausages so the NaCl content is higher too. The less quantity of water loosed by the organic sausages caused lower salt concentration.

Author Response

Dear Reviewer 2, 

Thank you for your useful comments and suggestions. Below please find our answers:

Q: The title should be wider in order to include all the aspects considered in the work.

A: The title was corrected according to suggestion (Lines 2-5).

Q: All the methods have appropriate and relevant references of other authors. The design and organization of the trial (considering the concepts of samples, repetitions, treatments, etc.)  should be deeply explained and the words systematically used.

A: The manuscript was corrected according to request (see Lines 92, 94-95, 111, 119, 121).

Detailed comments:

Q: Line 86 - Staphylococcus aureus replaced by Staphylococcus aureus

A: Corrected (Line 89).

Q: Line 88 to 90 - “At the end of the processing, sausages from each treatment group (n=6 per treatment group) were randomly selected for further analyses, vacuum packed and frozen at -20°C or -80°C (depending on the analysis)”. To explain this better, namely the number of sausages.

Q: Is necessary to explain how many sausages for each treatment group for each analysis.

Q: Line 107 - How many sausages?

A: All mentioned analyses were performed on 6 sausages per treatment (12 in total). Sentence was slightly modified (Lines 92, 94-95, 111, 119, 121).

Q: Line 117 - Why to use the descriptor (sensory evaluation) rancid? Maybe that kind of flavour is due to other causes like volatiles originating from spices and not really rancidity.

A: It is good hypothesis - however, the results obtained for chemical oxidation (TBARS, carbonyls) or specific volatile compounds (related to rancid) show an increase of rancidity related compounds in organic sausages, which supports the results obtained for sensorial rancidity. Moreover, volatiles from spices were markedly decreased.

Q: Line 162 - Instead of table 3 to show data of sensory evaluation I suggest a graphic representation in order to allow the reader to easier compare treatment results. The data can be plotted onto graphs, such as the spider plot.

A: As suggested, Table 3 was changed to spider plot (Figure 2).

Q: Line 231 “The observed differences in salt content between organic and conventional group are difficult to explain”. Maybe the explanation is the higher value of weight loss during the process for conventional sausages so the NaCl content is higher too. The less quantity of water loosed by the organic sausages caused lower salt concentration.

A: Actually, the NaCl is expressed on dry matter basis, so the loss of water cannot be the reason.

Best regards.

Reviewer 3 Report

Review on “Aromatic profile and physicochemical traits of dry-fermented sausages produced without nitrites using pork from Krškopolje pig reared in ecological and conventional husbandry”

Comments to the author

General comments: the authors objective was to develop a nitrite-free product (salami type of dry-fermented sausage) from Krškopolje pigs and to evaluate if and how the husbandry system (organic or conventional) affects physico-chemical properties, sensory traits and volatile profile of dry-fermented sausages. The authors refer that results demonstrated lower quality of dry-fermented sausages from organic raised pigs mainly due to their higher PUFA values and the fact that no anti-oxidant was used to prevent its effect on oxidation.

In my opinion, the use of organic farming and local breeds is an excellent idea since consumers are more and more concerned with health issues and the preservations of local patrimony is of great importance. But the conclusions of this paper discard the advantages of organic farming and the elimination of artificial additives. Maybe the authors should consider that there are already several published works with the use of natural antioxidants. Nevertheless, they are worth as conclusions regarding the work that has been done.

I think this is a very well designed and written paper, and only a few considerations must be registered before it can be published. The main issue is related to the findings that sausages produced from organic pork have “lower quality” which is a bit disappointing in the demand for quality in traditional products. How do you define quality?

Specific comments:

Tittle: You should include “sensory analysis” in the title

Simple summary:

Lines 20-23: The period “Results of this study demonstrated lower quality of dry-fermented…without additives with antioxidant capacity” is confusing. You should rewrite it. Dry-sausages from organic pork have lower quality (what are the standards of quality?) because of high levels of PUFA and (I am not sure I understand) because they are produced without antioxidants. Maybe you should link these two effects.

Abstract:

Line 26: include “physical” analysis

Keywords: include “physical-chemical and sensory characteristics”

Introduction: ok

Material and methods: ok

Results:

Table 1: In 1 kg DM, 593g and 599g of fat, it’s a lot of fat!!

In a scale of 0 -9 (if I understood right) the sensory differences, although significant, are not that big in what you consider bad sensory attributes for organic sausages. Maybe it was interesting to ask consumers if they detect those differences and if the differences were important considering the way both types of pigs were raised and all the implications.

Discussion:

Line 242: include “..be attributed to the differences..”

Lines 257-259: Rewrite because it’s confusing “A decrease of neutral… from organic pigs.”

Line 287: it is you that agree with the study with Iberian breed and not the opposite!! I mean, the present study agrees with the previous one.

Conclusions:

Is the water activity related to the higher level of unsaturation, and consequently to the disadvantage of using organic pork for the dry-fermented sausages production? Also, the softer texture is a sign of bad quality, why? Because they lack consistency to cut? I think the dry-fermented sausages are to be consumed with no heat treatment. If so, you should refer that.

Maybe the organic sausages need more time to be ready for consumption.

Based on the demand for products without artificial additives, there are several published studies on the use of natural antioxidants, which could benefit, according to the results obtained in this work, organically reared pigs.

You could improve your conclusions saying that more studies should be done trying to overcome the disadvantages of organic pork demonstrated in this study, given the consumers demand for more traditional, natural, healthier and sustainable products.

Author Response

Dear Reviewer 3, 

Thank you for your useful comments and suggestions. Below please find our answers:

Q: In my opinion, the use of organic farming and local breeds is an excellent idea since consumers are more and more concerned with health issues and the preservations of local patrimony is of great importance. But the conclusions of this paper discard the advantages of organic farming and the elimination of artificial additives. Maybe the authors should consider that there are already several published works with the use of natural antioxidants. Nevertheless, they are worth as conclusions regarding the work that has been done.

A: Considering that comment, the conclusions were partly rewritten (Lines 340-345).

Q: I think this is a very well designed and written paper, and only a few considerations must be registered before it can be published. The main issue is related to the findings that sausages produced from organic pork have “lower quality” which is a bit disappointing in the demand for quality in traditional products. How do you define quality?

A: We agree with the reviewer on that point, so we now tried to abstain from that kind of “subjective” view/conclusions. The parts of the abstract and the conclusions were thus rewritten (Lines 20, 340-341).

Specific comments:

Q: Tittle: You should include “sensory analysis” in the title

A: Included (Line 2).

Simple summary:

Q: Lines 20-23: The period “Results of this study demonstrated lower quality of dry-fermented…without additives with antioxidant capacity” is confusing. You should rewrite it. Dry-sausages from organic pork have lower quality (what are the standards of quality?) because of high levels of PUFA and (I am not sure I understand) because they are produced without antioxidants. Maybe you should link these two effects.

A: Rewritten (Lines 20-23). See also answer to the previous comments above.

Abstract:

Q: Line 26: include “physical” analysis

A: Corrected to physicochemical (Line 26).

Q: Keywords: include “physical-chemical and sensory characteristics”

A: Included (Line 37).

Results:

Q: Table 1: In 1 kg DM, 593g and 599g of fat, it’s a lot of fat!!

A: Yes, it is quite a lot of fat. Raw material originated from a local pig breed, which has likely higher inter- and intramuscular fat deposition. We followed standardly practiced 20% backfat inclusion in the formulation, but due to higher inter/intra muscular fat it resulted in so high fat %.

Q: In a scale of 0 -9 (if I understood right) the sensory differences, although significant, are not that big in what you consider bad sensory attributes for organic sausages. Maybe it was interesting to ask consumers if they detect those differences and if the differences were important considering the way both types of pigs were raised and all the implications.

A: As answered above, abstract and conclusions were changed in order to avoid subjective evaluation of the product (Lines 20, 340-347).

Discussion:

Q: Line 242: include “..be attributed to the differences..”

A: Corrected (Line 260).

Q: Lines 257-259: Rewrite because it’s confusing “A decrease of neutral… from organic pigs.”

A: Sentence rewritten (Lines 276-277).

Q: Line 287: it is you that agree with the study with Iberian breed and not the opposite!! I mean, the present study agrees with the previous one.

A: Sentence rewritten (Line 305).

Conclusions:

Q: Is the water activity related to the higher level of unsaturation, and consequently to the disadvantage of using organic pork for the dry-fermented sausages production? Also, the softer texture is a sign of bad quality, why? Because they lack consistency to cut? I think the dry-fermented sausages are to be consumed with no heat treatment. If so, you should refer that.

A: Yes, to our opinion the higher level of unsaturation was related to lower dehydration resulting in  higher aw and softer texture, but there was no problems with cutting consistency. However, we agree that it is not apriori “bad quality” – so in line with this and previous comments, conclusions and abstract were rewritten (Lines 20, 340-347).

Q: Maybe the organic sausages need more time to be ready for consumption.

A: This is difficult to say, also depends on how long. With the prolongation of the processing also oxidation and proteolysis could exceed acceptable levels.

Q: Based on the demand for products without artificial additives, there are several published studies on the use of natural antioxidants, which could benefit, according to the results obtained in this work, organically reared pigs. You could improve your conclusions saying that more studies should be done trying to overcome the disadvantages of organic pork demonstrated in this study, given the consumers demand for more traditional, natural, healthier and sustainable products.

A: Conclusions changed accordingly (Lines 346-347).

Best regards.
